# Does the Microenvironment Hold the Hidden Key for Functional Precision Medicine in Pancreatic Cancer?

**DOI:** 10.3390/cancers13102427

**Published:** 2021-05-17

**Authors:** John Kokkinos, Anya Jensen, George Sharbeen, Joshua A. McCarroll, David Goldstein, Koroush S. Haghighi, Phoebe A. Phillips

**Affiliations:** 1Pancreatic Cancer Translational Research Group, School of Medical Sciences, Faculty of Medicine & Health, Lowy Cancer Research Centre, UNSW Sydney, Sydney, NSW 2052, Australia; john.kokkinos@unsw.edu.au (J.K.); g.sharbeen@unsw.edu.au (G.S.); d.goldstein@unsw.edu.au (D.G.); 2Australian Centre for Nanomedicine, ARC Centre of Excellence in Convergent Bio-Nano Science and Technology, UNSW Sydney, Sydney, NSW 2052, Australia; jmccarroll@ccia.org.au; 3Children’s Cancer Institute, Lowy Cancer Research Centre, UNSW Sydney, Sydney, NSW 2052, Australia; ajensen@ccia.org.au; 4School of Women’s and Children’s Health, Faculty of Medicine & Health, UNSW Sydney, Sydney, NSW 2052, Australia; 5Prince of Wales Clinical School, Prince of Wales Hospital, UNSW Sydney, Sydney, NSW 2052, Australia; kshaghighi@bigpond.com

**Keywords:** pancreatic cancer, microenvironment, precision medicine, pre-clinical models

## Abstract

**Simple Summary:**

Pancreatic ductal adenocarcinoma (PDAC) is one of the deadliest of all major cancers. We now know that a ‘one-size-fits-all’ approach when treating PDAC patients does not work, so research has focused on developing personalised treatment strategies based on the biology of each patient’s tumour. However, most of these strategies ignore the presence of the characteristic fibrotic PDAC stroma which can act like a fortress to protect the tumour from chemotherapy and drive its aggressive nature. In this review, we summarise the role of this ‘fortress’ in protecting PDAC from chemotherapy and highlight the need for pre-clinical models that are informing precision medicine to reflect this hallmark feature of PDAC. We also propose the power of ex vivo whole-tissue culture models that mimic the tumour and its surrounding fortress to inform personalised treatment for PDAC.

**Abstract:**

Pancreatic ductal adenocarcinoma (PDAC) is one of the most lethal cancers and no significant improvement in patient survival has been seen in the past three decades. Treatment options are limited and selection of chemotherapy in the clinic is usually based on the performance status of a patient rather than the biology of their disease. In recent years, research has attempted to unlock a personalised treatment strategy by identifying actionable molecular targets in tumour cells or using preclinical models to predict the effectiveness of chemotherapy. However, these approaches rely on the biology of PDAC tumour cells only and ignore the importance of the microenvironment and fibrotic stroma. In this review, we highlight the importance of the microenvironment in driving the chemoresistant nature of PDAC and the need for preclinical models to mimic the complex multi-cellular microenvironment of PDAC in the precision medicine pipeline. We discuss the potential for ex vivo whole-tissue culture models to inform precision medicine and their role in developing novel therapeutic strategies that hit both tumour and stromal compartments in PDAC. Thus, we highlight the critical role of the tumour microenvironment that needs to be addressed before a precision medicine program for PDAC can be implemented.

## 1. Introduction

Pancreatic cancer is set to become the 2nd leading cause of cancer related deaths by 2025 [1]. Affecting approximately 459,000 people worldwide in 2018, it is accompanied by the lowest 5-year survival rates of all major adult cancers at just 9% [2,3]. Its main subtype, pancreatic ductal adenocarcinoma (PDAC), makes up approximately 85% of pancreatic malignancies [3]. Despite advances in our understanding of PDAC biology, our best chemotherapeutic regimens utilise a “one-size-fits-all” approach and only extend survival by months. A barrier to the success of current therapies is the presence of both intrinsic and extrinsic factors that drive chemoresistance in PDAC tumours. Moreover, therapies traditionally target cancer cells only and neglect the important role played by stromal cells in chemoresistance. These stromal cells, mainly consisting of cancer associated fibroblasts (CAFs), mediate chemoresistance by forming the dense fibrotic scar tissue that characterises PDAC [4]. This fibrosis creates a physical barrier for drug delivery by collapsing vasculature and impairing effective chemotherapeutic drug delivery to the tumour cells. Similarly, their bi-directional interaction with surrounding PDAC cells fosters a unique microenvironment that enables tumour progression, metastasis and chemoresistance to conventional treatment regimens [4].

Currently, patients with resectable PDAC make up approximately 15% of all cases [5]. Neoadjuvant chemotherapy may be administered with the goal of eliminating any micro-metastases prior to restaging and tumour resection [6]. Following surgery, patients are assessed and administered either modified FOLFIRINOX (5-fluorouracil, oxaliplatin, leucovorin and irinotecan), gemcitabine and capecitabine, gemcitabine alone, or 5-fluorouracil and folinic acid [6]. The therapy is chosen after an assessment of the patient’s performance status. For the remaining majority (~85%) of patients presenting with unresectable locally advanced or metastatic PDAC, treatment options are geared more towards alleviating symptoms and palliative care to improve quality of life [7,8]. In 2020, the American Society of Clinical Oncology (ASCO) guidelines for the treatment for metastatic PDAC highlighted the benefit of including an initial assessment for actionable genomic alterations that may guide first-line treatment [8]. Whilst this highlights a positive shift towards a more precision based approach to PDAC treatment, we need to continue to improve our understanding of the basic biology of PDAC to inform an effective personalised treatment approach for PDAC.

## 2. Current Trends in PDAC Precision Medicine

In the past decade, significant progress has been made in understanding the molecular biology of PDAC. In fact, genetic profiling has helped us identify different PDAC subtypes that may respond differently to chemotherapy. In 2011, Collison and colleagues [9] analysed the transcriptome of resected patient PDAC tissue and identified three PDAC subtypes that they termed classical, quasi-mesenchymal and exocrine-like. Interestingly, each subtype correlated with varying degrees of patient survival and sensitivity to chemotherapy in vitro [9]. This has been validated in larger patient cohorts [10,11,12] identifying different PDAC genetic profiles that correlate with patient clinical outcome, as reviewed recently [13].

These findings have inspired an effort to profile the genetic signatures of PDAC patient tumours and use this information to inform individual patient clinical care. However, rather than understanding the subtype that characterises a particular patient’s tumour, the focus has shifted to identify patients that harbour clinically actionable targets [14,15,16,17,18,19]. Using tissue obtained from biopsy of patients with locally advanced or metastatic PDAC, Aguirre and colleagues [20] identified specific gene expression signatures that might be therapeutically relevant. For example, 8% of patients had a BRCA mutation, suggesting they might respond well to platinum-based chemotherapy (e.g., FOLFIRNOX) or a poly-ADP polymerase (PARP) inhibitor such as olaparib [20]. In a recent double-blind, randomised, placebo-controlled phase III clinical trial of 3315 patients with metastatic PDAC, 7.5% had a germline BRCA mutation, and of these patients with BRCA mutation, maintenance olaparib treatment significantly improved progression free survival [21]. Given that most PDAC patients present with advanced disease where obtaining a surgical pathology sample is not possible, the study by Aguirre et al. [20] using biopsy material presents a leap forward to identify patients with clinically actionable targets, without excluding those patients with advanced or metastatic disease.

However, although the concept of treating individual patients based on a particular genetic alteration is exciting, a major limitation is that very few PDAC patients have genetic alterations that are therapeutically targetable. In the Know Your Tumour program, the largest PDAC precision medicine trial reported to date, actionable molecular alterations were identified in 26% of patients who received personalised molecular testing results [22]. Importantly, patients who received a matched therapy based on their molecular alterations had significantly improved overall survival compared to patients that did not received matched therapy [22]. This provides strong evidence that when an actionable molecular alteration is identified, matched therapy can have positive outcomes on patient survival. Despite these promising results, however, only 46 patients (2% of total) received matched therapy out of 1856 patients who were referred to the Know Your Tumour program [22]. This suggests that in addition to the low frequency of actionable molecular alterations in the PDAC patient population, there are further barriers that currently hinder such precision medicine programs from becoming part of routine clinical care. For example, a study at John Hopkins Hospital found that only 3 out of 92 patients (3%) received matched targeted therapy based on tumour next-generation sequencing [23]. This study identified that a significant barrier is the delay in requesting genomic profiling—it was reported that the medium time to order genomic profiling was 229 days [23].

Nonetheless, even if this logistical barrier is overcome, a major challenge that remains is the vast majority of PDAC patients that do not harbour an actionable molecular target. For precision medicine to be applicable to all patients with PDAC, research should focus on finding ways to predict the standard-of-care chemotherapy drugs that would benefit each individual patient rather than limiting the patient population to the few that have a targetable molecular alteration. Currently, selection of chemotherapy regimens in the clinical setting is not based on the biology of each patient’s disease but rather based on a patient’s performance status to prevent excessive side effects [6,7,8]. One approach to better predict the chemotherapy drugs that will be effective on each individual patient, is to use clinically relevant patient-derived models to rapidly screen drugs and assess their response; an approach termed ‘functional precision medicine’ [24].

## 3. Pre-Clinical Models in the Precision Medicine Pipeline

For a pre-clinical model to guide functional precision medicine for PDAC, a key requirement is that it can provide meaningful and actionable results in a short timeframe. Patient-derived xenografts (PDX) are being used to inform personalised treatment in other cancers [25], but the length of time (several months) required for their establishment makes them unsuitable for cancers such as PDAC that have such short patient survival [26]. The cost and large number of mice required to expand patient tumour tissue in mice hinders the feasibility of implementing PDXs in routine clinical care [26]. Furthermore, most PDX models involve subcutaneous implantation of patient tumour tissue into immune-compromised mice which is a poor representation of the tumour microenvironment and low take rates can bias more aggressive disease [26]. Recently, organoids have gained much traction as a rapid and efficient model of interpatient heterogeneity with potential to guide functional precision medicine for PDAC [27,28].

To establish tumour organoids, fresh human or mouse tumour tissue is minced, enzymatically or mechanically digested, embedded in a matrix, and supported by culture medium that contains growth factors that enrich for cells of epithelial origin [29]. The first human PDAC tumour organoid model was established in 2015 [30] and has since gained much attention in the PDAC drug discovery and precision medicine pipeline. Importantly, most studies have reported a high success rate of organoid establishment from patient surgical or biopsy tumour tissue [30,31,32,33], meaning organoids can be derived from patients with both resectable and unresectable disease. In addition, organoids have been extensively validated in their ability to maintain the genetic features and histological architecture of tumour cells throughout organoid culture [30,31,33].

Given the ability of tumour organoids to rapidly and efficiently model interpatient heterogeneity, several studies have begun to investigate whether drug sensitivity profiles can predict patient response to chemotherapy and inform personalised treatment. Many of these early studies were performed in colorectal cancer tumour organoids. As part of an elegant study by Vlachogiannis et al. [34], the investigators generated tumour organoids from two separate liver metastases of a colorectal cancer patient. One of these liver metastases grew rapidly after the patient was treated with TAS-102, the nucleoside analogue trifluridine combined with the thymidine phosphorylase inhibitor tiparacil, whereas the other liver lesion remained stable in size after treatment [34]. Interestingly, the organoids from the liver metastasis that remained stable had a more chemosensitive dose-response profile to TAS-102 treatment compared to the organoids from the liver metastasis that grew rapidly following treatment [34]. This presents early evidence that drug response in the tumour organoid model may mimic patient response in the clinic. A more recent study showed that tumour organoids derived from rectal cancer patients may predict response to chemotherapy and radiotherapy [35]. Organoids that continued to grow after 5-fluorouracil treatment were derived from patients who subsequently did not respond to 5-fluorouracil chemotherapy in the clinic [35]. Organoids that decreased in size following chemotherapy, in general, correlated with patients who responded to treatment [35]. Similar results were observed in organoids treated with radiotherapy [35]. However, it is important to note that there was a subset of patients (12/80 patients) who had rapid progression of their tumour following treatment, but their tumour organoids were observed to be sensitive to chemotherapy or radiotherapy [35]. The findings suggested that in some patients, there are additional factors that can drive chemoresistance that are not recapitulated in the tumour organoid model.

In PDAC, the potential of tumour organoids to predict patient response to chemotherapy has also been investigated. Tiriac and colleagues [36] generated dose response curves to standard of care chemotherapy drugs from both primary PDAC tumours and metastatic disease. The authors separated the area under the curve (AUC) values into tertiles and questioned whether these tertiles correlated with patient response in the clinic [36]. In the small number of patients (*n* = 9) where clinical chemotherapy response data was available, the ‘chemoresistant’ tertile correlated to patients who had rapid disease progression during treatment, while the ‘chemosensitive’ tertile correlated to patients who responded to treatment [36]. We eagerly await the results of larger cohort studies to reproduce these encouraging early findings. However, it is important to acknowledge here that although the AUC values from dose-response curves can be separated into tertiles, all the organoids showed a decrease in cell viability with increasing concentration of drug [36]. And although there was a spread in AUC values, there was no clear separation or clustering of AUC values that could be categorised as ‘responders’ or ‘non-responders’ [36]. This raises the same question discussed above—is there a missing link in the tumour organoid model that prevents organoids accurately identifying all patients with chemoresistant disease? Is this missing link perhaps the complex tumour microenvironment and fibrotic PDAC stroma that drives PDAC resistance to chemotherapy?

## 4. How the Microenvironment Influences PDAC Response to Treatment

The majority of a PDAC tumour (up to 90%) is made up of extracellular matrix (ECM) and stromal cells [37,38,39]. Combined with tumour cells, they make up a highly heterogeneous and complex network that fosters a unique microenvironment for tumour cell growth [37,38]. Importantly, clinical evidence has demonstrated that high levels of fibrosis correlate with poor patient survival in PDAC [40,41]. There is also a crosstalk network between the CAFs which produce this characteristic fibrosis and their neighbouring PDAC cells to promote tumour progression. Furthermore, another feature of PDAC is the immune suppressive microenvironment orchestrated by CAFs, T regulatory cells, tumour associated macrophages and myeloid-derived suppressive cells [42]. Recapitulating these hallmark features of PDAC when considering preclinical models for functional precision medicine is critical. Importantly, many of these features play key roles in directly or indirectly contributing towards the chemoresistant phenotype of PDAC (Figure 1) [43].

### 4.1. Extracellular Matrix

Excessive extracellular matrix (ECM), or fibrosis, can make up to 90% of PDAC tumour volume [37] and represents a major barrier to chemotherapy treatment. PDAC fibrosis is comprised of a variety of ECM proteins including collagen and hyaluronan and can be interspersed with nerves and blood vessels [43]. This dense ECM causes compression of tumour vessels, increased intratumoral fluid pressure, and physical hinderance of chemotherapy access [43,44]. This has been demonstrated by two independent studies whereby inhibition of CAF proliferation using sonic hedgehog pathway inhibitors in mouse PDAC tumours reduced fibrosis and increased vascular perfusion [44,45]. Similarly, anti-fibrotic therapy using the angiotensin II receptor inhibitor losartan improved nanoparticle delivery to mouse orthotopic PDAC tumours [46]. Another antifibrotic therapy, using enzymatic degradation of hyaluronan with PEGPH20, was shown to normalise tumour blood vessels, improve gemcitabine drug delivery, and increase survival in vivo [47]. Although losartan is currently under clinical evaluation for combination treatment in PDAC [48], both sonic hedgehog pathway inhibition [49] and PEGPH20 [50] are no longer being investigated for PDAC anti-stromal therapy, thus demonstrating that the challenge of improving patient outcome with stromal targeting agents is more complex than simply ablating CAFs or depleting ECM components. Nonetheless, these early studies provide important proof-of-principle evidence that the ECM is a key component of the PDAC microenvironment that can hinder drug access and contribute towards chemoresistance.

The ECM can also promote chemoresistance in PDAC independent of its role in compressing blood vessels and hindering drug access. A study by Dangi-Garimella and others [51] showed that in contrast to cells cultured in 2D, PDAC cells cultured in a 3D type I collagen matrix demonstrated chemoresistance to gemcitabine. Similarly, collagen, fibronectin, and laminin, key proteins that make up the PDAC ECM, have been shown to increase PDAC cell chemoresistance in vitro [52,53,54]. For example, PDAC cells cultured in the presence of laminin demonstrated increased FAK phosphorylation which in turn activated AKT signalling to promote chemoresistance [53]. In ovarian and breast cancer cells, hyaluronan has been shown to bind to CD44 on tumour cells to stimulate multidrug resistance gene 1 (MDR1) expression, a key defence mechanism against chemotherapy [55]. Hyaluronan can also promote AKT signalling in tumour cells, activating a cascade of pro-survival and anti-apoptotic mechanisms to protect against chemotherapy-induced death [56]. In addition to the physical cues in the tumour microenvironment, the dense fibrosis and collapsed blood vessels can generate hypoxia which in its own way can drive chemoresistance. For example, hypoxia promotes chemoresistance by inducing the epithelial to mesenchymal transition (EMT) phenotype in PDAC [57]. This is a result of hypoxia inducible factor 1α (HIF-1α) and NF-κB hyperexpression which can decrease the sensitivity of PDAC cells to gemcitabine [57]. Furthermore, the metabolic rewiring of PDAC cells induced by the nutrient deprived microenvironment can also directly stimulate chemoresistance [58]. Nutrient deprivation in PDAC can increase glycolysis, promote expression of MUC1 and activate pyrimidine synthesis which can all contribute towards chemoresistance [59,60,61].

### 4.2. Cancer-Associated Fibroblasts

PDAC fibrosis is generated by cancer-associated fibroblasts (CAFs) [37]. In health, these cells are normally quiescent and are activated to repair tissue damage. In PDAC, however, these cells are hijacked by signals from PDAC cells [37]. CAFs directly play a key role in driving tumour progression and inducing PDAC chemoresistance. Interestingly, CAFs have been shown to metabolise and inactivate gemcitabine, acting as scavengers to prevent gemcitabine toxicity in PDAC cells [62]. CAFs can also secrete a range of paracrine factors to promote chemoresistance in neighbouring PDAC cells. For instance, CAF activation produces large amounts of leukaemia inhibitory factor (LIF), a paracrine signal that activates the STAT3 pathway in PDAC cells [63]. Shi and others [63] showed that inhibiting LIF with a neutralising monoclonal antibody increased sensitivity to gemcitabine in vivo. Moreover, to protect against gemcitabine toxicity, CAFs have been shown to expel deoxcycytidine, a gemcitabine analogue that interferes with gemcitabine activation in PDAC cells [64]. In vitro, PDAC cells recorded up to a 100-fold increase in the IC50 of gemcitabine when incubated with conditioned media from CAFs [64]. CAFs also play a critical role in repressing an anti-tumour immune response which may explain why PDAC tumours are largely unresponsive to immunotherapy. The desmoplastic stroma and dense ECM can act as a physical barrier to cytotoxic immune cell migration, and CAFs can also directly inhibit the cytotoxic activity of immune cells in PDAC, as reviewed recently [42,65]. For example, CAFs release the T cell chemoattractant CXCL12 which diverts cytotoxic CD8+ T cells migration towards the stromal compartment and away from the tumour cells [66].

Interestingly, subpopulations of CAFs were recently identified which exhibit both transcriptional and functional diversity [67,68]. The rise in single cell RNA sequencing has allowed us to identify distinct populations of myofibroblast CAFs (myCAFs), inflammatory CAFs (iCAFs) and antigen presenting CAFs (apCAFs) [69,70]. Importantly, although there is no evidence yet describing the role of these CAF subtypes in influencing PDAC response to chemotherapy, it can be hypothesised that different CAF populations may have varying effects on PDAC cell chemoresistance. This highlights the need for patient-derived models to reflect the heterogeneous nature of the PDAC stroma when informing precision medicine.

### 4.3. Immune Cells

In addition to CAFs, the PDAC stroma also houses infiltrating immune cells. The immune landscape of the PDAC microenvironment, especially the involvement of tumour-associated macrophages (TAMs), can directly contribute towards the development of chemoresistance. For example, Mitchem and others [71] showed that gemcitabine treatment in PDAC can encourage recruitment of TAMs into PDAC tumours where they decreased the response to gemcitabine. Importantly, targeting TAMs in combination with gemcitabine synergistically decreased tumour growth in PDAC orthotopic tumours [71]. TAMs have also been shown to express cytidine deaminase, an enzyme responsible for converting gemcitabine to its inactive form, dFdU [72]. More recently, Halbrook and colleagues [73] identified that TAMs secrete deoxycytidine which competitively interferes with gemcitabine action in PDAC cells. Although it has not been investigated whether other immune cells directly stimulate chemoresistance in PDAC, the overall immune-suppressive landscape induced by the ECM, CAFs, T regulatory cells, and myeloid derived suppressor cells prevents the cytotoxic function of CD8+ T-cells and natural killer cells which is necessary to complement the cytotoxic effects of chemotherapy.

### 4.4. Extracellular Vesicles

Another factor that can drive chemoresistance in the PDAC tumour microenvironment is the presence of extracellular vesicles (EVs). Over the last 20 years, EVs have gained significant attention after their important role as mediators of cell-to-cell communication was discovered [74,75]. Released from both tumour and healthy cells, EVs were initially proposed to be merely ‘garbage bags’ for cells. However, we now know that they carry important molecular cargo such as protein, lipids, DNA, RNA, as well as proteomic remnants of their host cell [76]. The latter has also led to extensive work proposing their use as biomarkers for tumour growth and treatment efficiency. For instance, EVs have been shown to carry double stranded DNA for KRAS and p53 which are frequently mutated in PDAC [77]. Importantly, there is an increasing amount of evidence detailing the reciprocal exchange of EVs between stromal and tumour cells. CAF-secreted EVs can confer gemcitabine chemoresistance in PDAC cells by transfer of various miRNAs [78,79,80,81]. As an example, EVs released by CAFs contain high levels of miR-21 which were subsequently engulfed and expressed in PDAC cells [81]. miR-21 is a well-known predictor of gemcitabine chemosensitivity in PDAC and promotes a shift to an EMT phenotype in recipient PDAC cells [81,82]. Contrastingly, there is also evidence alluding to a tumour suppressive role of stromal derived EVs in PDAC. Han and others [83] describe the transfer of miR-145 from the stroma to recipient PDAC cells, decreasing cell viability and inducing cell death. This reinforces the need for clinically relevant models that accurately mimic the PDAC tumour microenvironment and the importance of treating individual patients based on the biology of their disease. Currently, the representation of EVs in current 3D models is lacking and may be contributing to the variable results seen in current models. An important study addressing these issues confirmed that miRNA in PDAC-derived EVs is altered depending on the culture method [84]. Zeöld and others [84] concluded that PDAC cells grown as organoids had the greatest overlap in miRNA profiles when tested against miRNA from blood plasma samples from patients.

EVs are not only released from stromal cells but can also be secreted from PDAC cells to increase chemoresistance. For example, EVs released from gemcitabine treated PDAC cells expressed significantly higher levels of miR-155 which is laterally transferred from EVs to recipient tumour cells [85]. These recipient cells exhibited lower levels of deoxycytidine kinase—an enzyme necessary for the intracellular activation of gemcitabine. Combining an anti-miR-155 approach restored deoxycytidine kinase levels and improved sensitivity to gemcitabine [85]. Moreover, the EVs secreted by gemcitabine-treated PDAC cells led to upregulation of both CAT and SOD2, which are reactive oxygen species (ROS) detoxifying enzymes [85]. Maintaining low ROS levels protects PDAC cells from gemcitabine.

Overall, the above evidence demonstrates the critical role of the tumour microenvironment in driving the chemoresistant nature of PDAC. If we are to use patient-derived models to inform personalised treatment and predict patient response to therapy, the models that we use must reflect the features of PDAC that are known to drive chemoresistance and metastatic spread.

## 5. Models That Reflect the Complex Microenvironment Are Required to Inform Functional Precision Medicine in PDAC

Despite the many advantages of the tumour organoid model, many of the microenvironmental features discussed above are not recapitulated. Several studies have recently attempted to co-culture organoids with CAFs or immune cells, as reviewed recently [27,28], but the CAFs added to the organoids are not matched to the same patient and these co-culture systems also lack the presence of an ECM that matches human disease.

Several independent studies have now sought to bridge this gap using whole tumour tissue slices or tissue explants ex vivo (Figure 2) [86,87,88,89,90,91].

### 5.1. Ex Vivo Tissue Slice Model

One of the first studies to culture PDAC tissue slices was published in 2009, where surgical PDAC tissue was cut into slices and cultured for 6 days submerged in growth media [92]. Since then, other studies have performed more robust characterisation and validation of the ex vivo slice model. A team from the University of Washington showed that human surgical PDAC slices can be cultured for up to 9 days on membrane culture inserts [86,88]. Importantly, the study demonstrated that tumour and stroma architecture was maintained, as well as the presence of T cells and macrophages [86,88]. Misra et al. [87] performed a more detailed characterisation of the ex vivo slice model, showing that slices can be maintained for 96 h with the presence of tumour cells, CAFs and immune cells. A follow-up study by the same team showed that the transcriptome of these slices is largely maintained throughout the culture [90]. Roife and colleagues [89] have since shown that drug sensitivity to gemcitabine and irinotecan in tissue slices from PDAC patient-derived xenograft models was consistent with patient response in the clinic, suggesting the model could potentially inform personalised therapy.

### 5.2. Ex Vivo Tumour Explant Model 

Our lab has recently published a similar model where we cultured whole tissue explants (1–2 mm diameter) from surgical PDAC tissue for 12 days on haemostatic gelatin sponges [91]. We sought to establish this model as a less technically challenging method to culture patient PDAC tissue compared to slice cultures which require tissue to be dissected into thin slices of approximately 350 μm thickness [87]. This model represents a major step forward as it allows us to culture patient PDAC tumour tissue for a longer duration compared to slice cultures, and we also performed a comprehensive characterisation of the patient explants throughout culture. We showed that tumour cells and CAFs remain viable and maintain their 3-dimensional architecture and fibrosis throughout the 12-day culture window [91]. Importantly, by demonstrating bromodeoxyuridine uptake at day 12, we showed that both tumour and stromal cells have *de novo* cell proliferation and are negative for a cell death marker, TUNEL [91]. Although immune cells are a scarcity in PDAC tumours, we showed that when CD45+ lymphocytes are present in the patient’s tumour, they are maintained for at least 5 days in our explant culture [91]. As proof-of-principle, we tested abraxane chemotherapy in the explant model, and observed both responders and non-responders [91]. Interestingly, a patient whose explants were non-responsive to abraxane had recurrence of their primary PDAC tumour following gemcitabine and abraxane chemotherapy, providing early evidence that drug response in the tumour explant model may mimic patient response in the clinic [91]. In ovarian cancer, a larger study of 22 patients used the same gelatin sponge tumour explant model to demonstrate that explant response to carboplatin matched patient response in the clinic [93]. Importantly, dosing of chemotherapy in the explant model can be selected to match the concentration of drug in circulation, and the drug reaches the explants via capillary action through the gelatin matrix rather than artificially bathing the tumour cells as occurs in tumour organoids. Furthermore, we demonstrated the ability to deliver gene-therapy siRNA drugs using nanomedicine to patient-derived tumour explants *ex vivo*, raising the potential to silence undruggable genes in a preclinical model on a patient-by-patient basis [91]. The presence of the microenvironment and stroma in the tumour explant model represents a powerful tool to study the effects of gene-therapeutic nanoparticles on both tumour and stroma [94]. We recently demonstrated the utility of the tumour explant model to test the therapeutic effects of inhibiting SLC7A11 using either an siRNA-nanomedicine or a clinical-grade SLC7A11 inhibitor, sulfasalazine [95]. We demonstrated anti-tumour and stromal CAF reprogramming [95].

### 5.3. Opportunities of Whole-Tissue Models to Guide Functional Precision Medicine

The ability to culture patient PDAC tumour tissue ex vivo without manipulating its multicellular architecture presents exciting new opportunities for the precision medicine pipeline. Although we are yet to see concrete evidence in large patient cohorts that these whole-tissue models can accurately predict patient response to chemotherapy, we can at least hypothesise that their ability to model the complex microenvironment of human PDAC will enable them to more precisely predict patient chemotherapy response compared to other models that study tumour cells alone. While tumour organoid models often use IC50 or AUC values to classify ‘responders’ or ‘non-responders’, whole-tissue models have the potential to study the histological effects of a drug on tumour and stromal cells and could further classify response to drug as ‘complete response’, ‘partial response’, ‘stable disease’, or ‘progressive disease’. Another advantage of whole-tissue models is the rapid timeframe to generate a clinically meaningful result. Pharmacotyping in PDAC tumour organoids has been recently shown to be achievable after a median of 48 days [96], whereas our PDAC tumour explant model allows a result to be obtained shortly after the 12-day culture window [91]. This does not negate the use of organoids but reinforces the need to better understand the advantages and limitations of each preclinical model and select models based on their strengths. For example, in contrast to whole-tissue models, tumour organoid models have the ability to screen a broad range of drugs in a high-throughput setting. Thus, an ideal precision medicine pipeline could involve a combination of both tumour organoids and tumour slice/explant from the same patient, cultured in parallel, to draw from the strengths of each model. This could be even further strengthened by isolating tumour cells and CAFs from the same patient’s tumour, allowing a rich biobank to be formed of whole-tissue cultures, tumour organoids, and isolated tumour cells and CAFs.

For patients with surgically resectable PDAC, tumour organoids can be established from biopsy material when/if a diagnostic biopsy is performed (Figure 3). Organoid culture in the pre-operative setting could allow for tumour genomic profiling to potentially identify any targetable molecular alterations, and pharmacotyping to occur in parallel to select effective chemotherapeutic regimens. For this to be effective however, the median of 48 days previously reported [96] from tissue collection to obtaining the pharmacotyping result will need to be significantly shortened. Following this, whole-tissue ex vivo models could be established from tissue obtained during surgical resection. This would allow for any potential hits identified in the organoid genotyping or pharmacotyping to be validated and to select an ideal adjuvant chemotherapy regimen.

For patients with unresectable or metastatic PDAC, future research should attempt to culture explants/slices from biopsy material so that such a pipeline could accommodate the entire PDAC patient population. The challenge here, however, is that biopsy material is usually limited, and most tissue needs to be sent to pathology for diagnostic purposes. Even if biopsy tissue can be set aside to establish tissue explant culture, the small size of the biopsy material would limit the number of tissue explants that can be cultured from each patient, and hence would limit the number of chemotherapeutics to be tested. To overcome this challenge, active collaboration is required between researchers, gastroenterologists, and interventional radiologists to ensure that enough tissue from metastatic PDAC patients can be used for research purposes, without compromising the importance of pathology diagnosis. This may involve consent for more than one biopsy site at the time of the procedure.

However, even if biopsy of metastatic PDAC does not yield enough tissue to help select patient treatment, there are other avenues by which the tumour explant model could be used to culture metastatic PDAC. The tumour explant model could be adapted for use with a rapid autopsy program whereby both the primary tumour and multiple metastatic lesions could be cultured in parallel. Although this would not be suitable in an interventional precision medicine setting, it could be used to retrospectively uncover mechanisms of chemoresistance and to better study PDAC tumour biology [97]. This could potentially identify certain immunohistochemistry signatures associated with chemoresistance that could guide interventional trials in the future. Similarly, tumour explants could also be established from PDX models to culture the primary tumour and metastatic nodules and assess their response to treatment.

Importantly, such models could become powerful tools to study the dynamic biology and microenvironment of metastatic PDAC which is rarely studied using traditional in vivo or in vitro models. For example, a recent study elegantly identified distinct tumour promoting and tumour repressing functions of CAF subtypes in mouse PDAC liver metastases [98]. The tumour explant model, if successful in culturing metastatic PDAC, has potential to further investigate these questions in a fully human and dynamic culture system.

Nonetheless, the next step is to determine whether whole-tissue explant cultures can predict patient response to chemotherapy. For this to be determined, a non-interventional trial is first needed to assess the response to standard of care adjuvant chemotherapy drugs in the tissue explant model, and then retrospectively assess how this correlates with subsequent patient response in the clinic. This could then be followed by an interventional clinical trial where the tumour explant model is used to generate drug sensitivity profiles to standard of care drugs which is then used to guide selection of treatment. The initial non-interventional trial would also be useful to determine the most accurate and efficient readout for drug response in the tissue explant model. We showed that terminal deoxynucleotidyl transferase dUTP nick end labeling (TUNEL) staining, as a measure of cell-death, can separate responders and non-responders to abraxane in a small number of patients [91], but this should be further evaluated in a larger patient cohort and testing a broader range of chemotherapeutics. Although an endpoint assessment of cell death or cell proliferation in response to a drug using the tissue explant model may provide clinically meaningful results, it would be helpful to develop a real-time method to assess explant response to treatment. This would allow us to observe how tumour and stromal tissue respond to chemotherapy over time rather than at a fixed snapshot. This could potentially involve live tissue imaging techniques or an analysis of secreted factors (e.g., DNA, metabolites, EVs) that are released into the medium reservoir over time.

### 5.4. Targeting the PDAC Stroma and Opportunities for Personalised Treatment

The presence and abundance of human and patient-matched stroma in whole-tissue ex vivo models can become a powerful tool to better understand the biology of the stroma in PDAC and the potential to target the PDAC stroma in a personalised setting. The concept of targeting the stroma in PDAC remains controversial given the debated role of the stroma in promoting or restraining tumour progression. However, many of the initial studies that instigated this controversy drew conclusions from mouse models that genetically ablated either α-smooth muscle actin expressing cells [99] or sonic hedgehog signalling [100,101] in PDAC cells where an increase in tumour metastasis was observed. However, it needs to be understood here that such approaches would also impair blood vessels, potentially increasing hypoxia which is known to promote PDAC progression. Furthermore, ablation of CAFs or inhibition of CAF signalling from the earliest stage of PDAC tumour development does not represent a therapeutic approach and differs substantially from novel therapeutic strategies that aim to inhibit the known tumour-promoting properties of CAFs. Nevertheless, recent evidence implies that targeting the stroma alone is unlikely to be clinically effective, and the right patients need to be selected for stromal targeting strategies. For example, an enzymatic approach to degrade hyaluronic acid (a key component of PDAC fibrosis) using PEGPH20 was shown to have promising pre-clinical efficacy in PDAC mouse models [47], but the recent HALO-301 phase III clinical trial of PEGPH20 combined with gemcitabine and abraxane in metastatic PDAC patients was terminated early as no increase in patient survival was observed [50]. Importantly, hyaluronic acid is only one component of the PDAC stroma, and enzymatic degradation of hyaluronic acid would have left untouched the CAFs that drive the fibrosis and pro-tumour signalling. In addition, eligibility for HALO-301 required patients to have high expression of hyaluronic acid, but for the majority of patients, hyaluronic acid expression was measured from biopsies obtained from metastatic sites [50]. Currently, it remains unknown whether the stromal signature of PDAC metastases is comparable to its primary tumour, thus suggesting the possibility that the wrong patients were selected for the HALO-301 trial.

Taken together, these landmark studies reinforce the need for preclinical models to accurately reflect the stromal and microenvironmental features of human PDAC to be able to guide stromal targeting strategies and to help select ideal patients. Despite failed clinical trials, research has focussed more on stromal reprogramming rather than stromal ablation [102]. There are many different approaches to therapeutically reprogram the PDAC stroma including inhibition of the tumour promoting properties of the ECM, normalising tumour vasculature, reverting CAFs to a quiescent/inactive state and inhibiting the pro-tumour properties of CAFs [102,103]. Adding to this complexity is the recent evidence showing prominent heterogeneity in the PDAC stroma [67,68] which implies that individual patients may respond differently to various stromal targeting agents. This highlights the need for stroma-rich patient-derived models to be able to test the functional effects of stromal targeting on a patient-by-patient basis. Furthermore, given that stromal targeting by itself is unlikely to be effective, dual therapeutic strategies that hit both tumour and stromal compartments represent an attractive area for PDAC treatment. For such therapeutic strategies to be evaluated, preclinical models that contain matched patient tumour and stroma would be highly beneficial and potentially even necessary. We recently demonstrated the utility of the ex vivo tumour explant model to test therapeutic inhibition of an amino acid transporter SLC7A11 using an siRNA nanoparticle and a clinical-grade pharmacological inhibitor, sulfasalazine [95]. Importantly, upregulation of SLC7A11 in stromal cells of PDAC correlates with poor patient survival [95]. Silencing of SLC7A11 expression using our siRNA nanoparticle in patient derived PDAC explants demonstrated potent anti-tumour and anti-stromal effects, and similar results were observed with sulfasalazine treatment of the patient explants [95]. This reinforces the ability of the ex vivo whole-tissue explant model to test such therapeutic strategies, and their potential to rapidly model tumour and stromal function can make them key components of novel clinical trial design to select patients that would benefit from such strategies.

## 6. Conclusions

For the first time in more than three decades, our improved understanding of the genetic landscape, the multicellular 3-dimensional architecture, and the hostile microenvironment of PDAC has potential to revolutionise the dismal survival of PDAC patients. But as momentum builds in the field of genetic profiling of PDAC tumours with a view towards precision medicine, we must not forget the critical role of the tumour microenvironment and the fibrotic stroma in driving the aggressive nature of PDAC. The advent of new culture techniques that allow us to grow whole pieces of human PDAC tumours ex vivo can re-unite these two fields by testing drug efficacy on each patient’s tumour in models that accurately mimic the microenvironmental cues and the tumour and stromal organisation in the same 3-dimensional architecture as present in the patient’s disease. This also paves the way for the development of a new class of drugs that therapeutically target both tumour and stromal compartments in PDAC.

Such research provides an ideal translational cycle of bench-to-bedside then back-to-the-bench. Using clinically relevant patient-derived models to predict response to chemotherapeutics can inform clinical decision making, and at the same time, testing drugs in models that contain the key features and complexities of human PDAC can provide a unique insight into the mechanisms by which these drugs work and the broader functional biology of tumour and stroma in PDAC. Although the fruits of precision medicine are yet to be truly reaped for PDAC patients, this new era offers great hope to the patient community but at the same time, reinforces the need to consider all aspects of PDAC biology while we launch this trajectory of precision medicine.

## Figures and Tables

**Figure 1 cancers-13-02427-f001:**
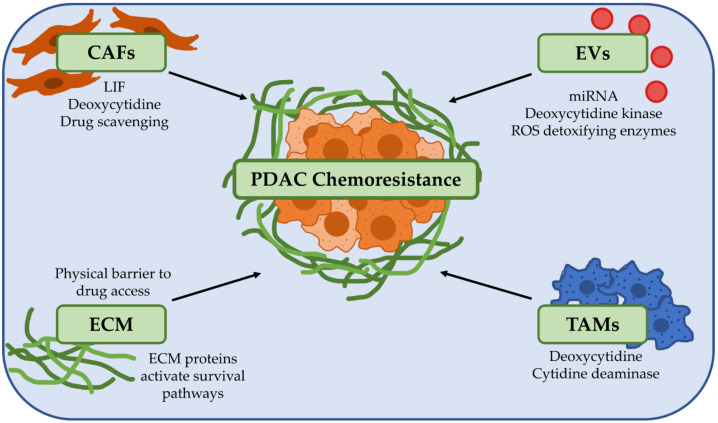
The tumour microenvironment is a key player in promoting chemoresistance in pancreatic ductal adenocarcinoma (PDAC). PDAC chemoresistance can be influenced by a broad range of factors in the tumour microenvironment. Cancer-associated fibroblasts (CAFs) secrete factors such as leukaemia inhibitory factor (LIF) and deoxycytidine which directly promote drug resistance and can also metabolise and inactivate drugs such as gemcitabine. Extracellular vesicles (EVs) and tumour associated macrophages (TAMs) can also secrete different factors to drive chemoresistance in PDAC cells. The extracellular matrix (ECM) can be a physical barrier to drug delivery by compressing blood vessels, and proteins that make up the ECM such as collagen, laminin and hyaluronan can activate survival pathways in PDAC cells to increase chemoresistance. Abbreviations: CAFs: Cancer-associated fibroblasts; ECM: Extracellular matrix; EVs: Extracellular vesicles; LIF: Leukaemia inhibitory factor; PDAC: Pancreatic ductal adenocarcinoma; ROS: Reactive oxygen species; TAMs: Tumour-associated macrophages.

**Figure 2 cancers-13-02427-f002:**
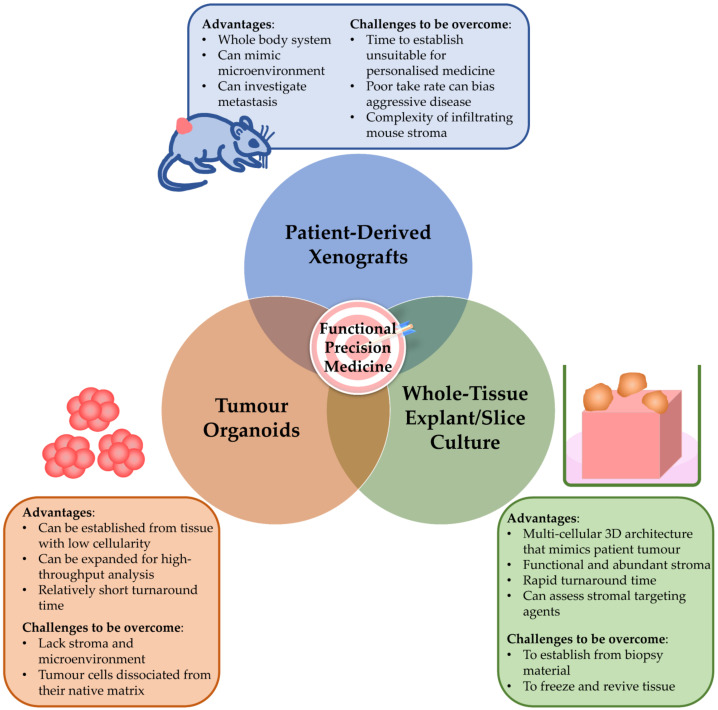
Pre-clinical models to inform functional precision medicine in pancreatic ductal adenocarcinoma. Patient-derived pre-clinical models are needed to inform functional precision medicine. Importantly, each model has its strengths and limitations, and these need to be addressed when choosing models to guide precision medicine.

**Figure 3 cancers-13-02427-f003:**
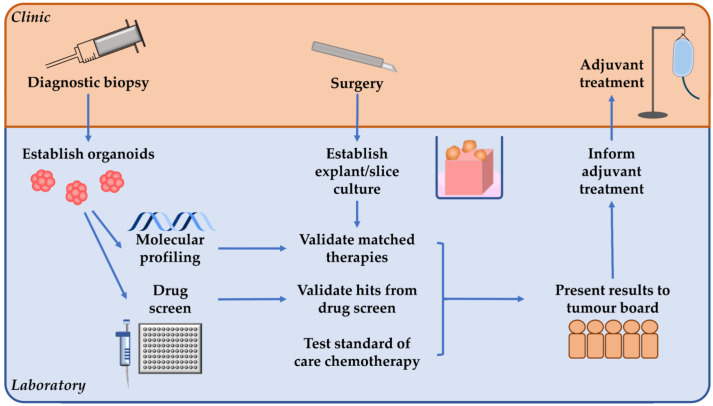
A proposed personalised medicine pipeline for patients with surgically resectable pancreatic ductal adenocarcinoma. If diagnostic biopsy is performed prior to surgery, organoids can be established to allow molecular profiling and drug screens to be performed. Then tumour explant/slice culture can be established from tissue obtained during surgery, which could allow any hits identified from the organoid molecular profiling or drug screen to be validated. Ex vivo explant/slice culture can also be used to test standard of care chemotherapy which can be particularly useful when no matched therapies are identified. These results can then be presented to a tumour board to guide personalised adjuvant treatment.

## Data Availability

Not applicable.

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
