# Peer review of "Does the Microenvironment Hold the Hidden Key for Functional Precision Medicine in Pancreatic Cancer?"

_cancers, 2021, doi:10.3390/cancers13102427_

Round 1
Reviewer 1 Report
This is a well written review on the subject of pancreatic cancer modeling platforms being utilized to guide selection of therapy. The focus is on the role of the tumor microenvironment (TME) and its contribution to drug resistance. The review is comprehensive, cites important recent literature and fairly describes the strengths and weaknesses of various models. The authors touch briefly on the question of metastatic disease and the question of how the TME may differ from that of the primary. Given most patients present with metastatic disease, it would be helpful to clarify the tissue requirements for tumor explant/slice culture and the limitations in testing multiple therapies using this method. I would also be interested in the authors perspective on how best to validate these preclinical models as truly clinical useful- ie) what types of co-clinical trial design might they feel would be informative to provide sufficient evidence for adoption. Finally, a comment on the actual efficacy endpoints used in these models would helpful- should we be measuring proliferation, cell death, changes in metabolism- a combination? Overall, a well written paper summarizing the state of the field.
Reviewer 2 Report
The article from Kokkinos and colleagues discusses the role of the tumor microenvironment on chemotherapy and the need to integrate the tumor microenvironment into precision medicine pipelines for pancreatic cancer. The article is very informative, well written and includes the most recent and relevant literature. The structure of the article is excellent and guides the reader through an interesting and important topic. I only have two minor comments: (1) there is a spelling mistake on page 2, line 63 for 5-Fluorouracil. (2) Page 7, line 224, I thought the first study using sonic hedgehog pathway inhibitors was published from Kenneth Olive and colleagues in 2009 in Science.
I congratulate the authors for this wonderful review.
